# DNA Barcoding Study of Representative *Thymus* Species in Bulgaria

**DOI:** 10.3390/plants11030270

**Published:** 2022-01-20

**Authors:** Ina Aneva, Petar Zhelev, Georgi Bonchev, Irina Boycheva, Stiliana Simeonova, Denitsa Kancheva

**Affiliations:** 1Institute of Biodiversity and Ecosystem Research, Bulgarian Academy of Sciences, 1113 Sofia, Bulgaria; seven_u@abv.bg; 2Department of Dendrology, University of Forestry, 1797 Sofia, Bulgaria; petar.zhelev@ltu.bg; 3Institute of Plant Physiology and Genetics, Bulgarian Academy of Sciences, 1113 Sofia, Bulgaria; georgi.bonchev71@gmail.com (G.B.); irina_boycheva@bio21.bas.bg (I.B.); steli_sim@abv.bg (S.S.)

**Keywords:** genetic markers, taxonomy, medicinal plants, phylogeny, taxonomically complex groups (TCGs)

## Abstract

We present a study on the taxonomy of eleven *Thymus* species, belonging to two sections and occurring naturally in Bulgaria. Four DNA barcoding markers—matK, rbcL, trnH-psbA and ITS—were applied to discriminate the species and to reveal their phylogenetic relationships. The results showed that rbcL has the lowest discriminating power regarding the studied species, while the other markers yielded results fitting better to the existing taxonomic schemes based on morphological traits. However, even in the case of better performing markers, the results were not straightforward—morphologically distinct species belonging to different sections were grouped together, and closely related species appeared genetically distinct. The results are typical for taxonomically complex groups, such as the genus *Thymus*, characterized in Bulgaria with great diversity, high percentage of endemism and still requiring a full and comprehensive taxonomic study. The results are discussed in the light of unresolved taxonomic problems and application of DNA barcoding methods.

## 1. Introduction

Resolving the problems arising when studying taxonomically complex groups (TCGs) requires a combined approach consisting of classical (morphological, anatomical, cytological) and modern (molecular) methods. Representatives of the genus *Thymus* can be a good example of a complex group encompassing many taxa, some of them with uncertain status, related among each other by hybridization, overlapping phenotypic variation and other attributes of the reticulate evolution, making the task of taxonomists more difficult [1].

The complex systematics of the genus *Thymus* has been outlined in many studies attempting to resolve the puzzle or a part of it [2,3,4,5,6]. Most of the challenges still stand today, and in many cases, the application of modern molecular methods did not provide a clear solution to taxonomic problems [1,7].

Currently, the number of species of the genus *Thymus* in Bulgaria is 21 [8,9,10,11], and the species list slightly differs from the one in the Euro+Med PlantBase (https://ww2.bgbm.org/EuroPlusMed/; last accessed 23 December 2021). In terms of species diversity, Bulgaria is among the richest countries in Europe (see also [12], for review). The genus *Thymus* is subdivided in two subgenera: *Coridothymus* (Reichenb. f.) Borbás and *Thymus* [2]. All Bulgarian species belong to the nominate subgenus.

Due to the importance of *Thymus* species as medicinal and aromatic plants and because of the conservation value of many species, they always provoked a substantial interest and have been subjected to diverse studies.

The last comprehensive study on the systematics of genus *Thymus* in Bulgaria was published more than 30 years ago [8], and practically no other taxonomic studies have been performed afterwards, with the exception of some floristic notes [12] and few studies on the essential oil composition [13,14]. Recently, Sostarić et al. [15] studied the genetic diversity and relationships among seven species of section *Serpyllum* (Miller) Bentham from Serbia. Apart from the taxonomic studies, essential oils and some other bioactive compounds of the species have provoked substantial interest [16,17,18,19,20].

Genetic diversity, evolution and phylogeny of *Thymus* species have received considerable attention [21,22,23], with application of newly developed methods. DNA barcoding is one of the modern methods applied successfully to the taxonomy of various groups of living organisms [24,25,26,27]. It is often pointed out that in the plant application of barcoding markers, distinguishing among the species is more difficult and not as straightforward as in animals and other organismic groups. The application of barcoding to TCGs in plants experiences some marked difficulties. The success of choosing an appropriate marker that could differentiate between closely related and morphologically similar species depends on many factors, among them polyploidy, the degree of relatedness of taxa of interest, gene flow and hybridization, dispersal ability and other life-history traits (see [25]). Several large-scale phylogenetic studies were performed at a higher than species level and revealed the phylogeny of the genera and tribes within Lamiaceae [28,29,30]. Genus *Thymus* was found to be paraphytletic to *Argantoniella* and *Saccocalyx* in both nuclear and plastid markers and to *Origanum* in a plastid marker only [28] and was placed within a clade together with *Thymbra*, *Origanum*, *Satureja* and *Micromeria* [30]. DNA barcoding has been successfully applied for identifying different species in commercial samples of herbs [31] and for the identification of different Lamiaceae species [32]. However, its application to infrageneric *Thymus* taxonomy did not allow definite conclusions [1,28,33,34]. Evidently, there are many unresolved problems related to the application of DNA barcoding to TCGs, but the approach is promising and will surely be improved further [26,27,35]. Therefore, it is worthy of applying this class of markers to a TCG, whose representatives were studied to a lesser extent.

The objective of the present study was to apply DNA barcoding markers to a representative set of the Bulgarian species of the genus *Thymus* in order to reveal the relationships among the species in the taxonomic scheme of the genus and to test the effectiveness of the markers for the study of *Thymus* taxonomy.

## 2. Results and Discussion

### 2.1. Efficiency of PCR Amplification and Sequencing

The success rates for PCR amplification and sequence efficiency were measured for all DNA barcodes obtained using the respective primers (Table 1). In the genus *Thymus*, primers used for different barcodes showed 100% amplification and sequencing efficiency among the 15 tested samples. One sample did not amplify, and one sample failed to be sequenced for ITS primers. Alignment length was 760 bp for matK, 530 bp for rbcL, 350 for trnH-psbA and 619 for ITS.

### 2.2. Genetic Diversity of Thymus Species and Accessions

Table 2 represents the parameters of genetic diversity of the studied DNA barcode regions. The total number of sites varied between 351 (trnH-psbA) to 761 (matK) and up to 1290 when combinations of different barcodes were considered. However, in all cases, more than 90% of the total number of sites were constant. The number of variable sites varied from 2 to 16 per region and increased to 29 in the combinations of two regions. The number of parsimony-informative sites was of similar magnitude and varied from 2 to 13.

We used the software package Geneious to construct a phylogenetic tree to infer genetic distances and the taxonomic relationship between *Thymus* accessions. A test of different genetic distance models (see Material and Methods) available in the package was performed under the clustering method UPGMA (unweighted pair group method with averages). Among the three models, Jukes–Cantor and Tamura–Nei models displayed highly comparable patterns of clustering of *Thymus* accessions for all analyzed DNA barcode regions and thus were both considered relevant for use (Appendix A). For further analyses, we used the Jukes–Cantor model, and the constructed phylogenetic trees for different analyzed DNA barcode regions are presented in Figure 1.

The level of genetic discrimination of *Thymus* specimens based on genetic distances differed between DNA barcode regions used. The rbcL region showed the lowest level of genetic differentiation, with the species specimens *T. stojanovii*, *T. thracicus* and *T. pulegioides* splitting into a distinct cluster (Figure 1a). The data from the rbcL region reflect the close genetic relationships of these species (Figure 1b). While the three species of this cluster belong to the same section *Serpyllum* (*T. pulegioides* to subsection *Alternantes* and the other two species to subsection *Pseudomarginati*), the second cluster includes all remaining species, which belong to the two sections (*Serpyllum* and *Hyphodromi*). The sections were specified based entirely on morphological characters, and some differences between the classification based on morphology and that based on genetic markers are expected. However, the differences and grouping based on the rbcL barcode region do not show some particular trend. Therefore, we consider this region to have little information value for the taxonomic classification of *Thymus*. Federici et al. [1] found the same sequence length of rbcL in all species studied, and the overall K2P distance was the lowest of all barcoding markers (0.1%).

The other region (trnH-psbA) showed the highest level of genetic divergence (Figure 1c). Three main groups were formed—two small and a bigger one. The first small group consisted of three species belonging to section *Hyphodromi*, and the second one combined species of section *Serpyllum*—similarly to rbcL tree topology. The larger group consisted of three species of section *Serpyllum* and two of section *Hyphodromi*. One species—*T. vandasii*—had a somewhat distinct position.

No particular trend could be observed in the third DNA barcode region matK. Several micro-clusters were formed combining species belonging to different sections, and different accessions of the same species were grouped in different clusters.

The fourth DNA barcode region—ITS—yielded a construction consisting of three clusters and one species distant from the others (Figure 1d). Again, like in the trnH-psbA region, this species was *T. vandasii*. The clusters combined species belonging to different sections—for example, the first small cluster consisted of *T. zygioides* (sect. *Hyphodromi*) and two accessions of *T. stojanovii* (sect. *Serpyllum*). However, it can be noted that here, different accessions of the same species clustered together, contrary to the other barcode regions.

The success of DNA barcoding in distinguishing taxa at the species level in plants depends on many factors [25]. It differs among the different groups and is usually lower in the TCGs, such as genus *Thymus*.

It has been reported in many studies that DNA barcoding leads to 90% success in species identification and differentiation [36,37,38]. However, there were also reports of lower success, especially in the TCGs. For example, [7] obtained about 60% success in species identification and delimitation in sedges (*Carex* spp.).

It is known that the combination of DNA barcode markers can improve the resolution of taxonomic and evolutionary studies. Therefore, here, we made an attempt to find out whether a combination of DNA barcode markers that have shown little information value can improve the taxonomic classification of the studied taxa. Phylogenetic trees based on the combination of plastid markers are shown in Figure 2.

Overall, based on data from the four DNA barcode regions, we can conclude about the presence of pronounced genetic diversity within the genus *Thymus*. The analyzed rbcL and matK regions alone cannot be used for relevant taxonomic differentiation of *Thymus* species and accessions. The most effective in distinguishing species and grouping closely related taxa of *Thymus* together was the ITS region. It was also the most informative region for other TCGs, such as the family Meliaceae [39] and a large set of medicinal plants [38]. It was reported by [39] species belonging to the same TCGs often have identical sequences of cytoplasmic DNA barcoding regions which greatly reduces the discrimination power of these markers. It is often recommended to use two-loci and/or multilocus combinations for obtaining better results [35,40]. However, in many studies using multulocus combination of DNA barcoding, markers did not substantially improve the resolution and identification power [39,41].

Phylogenetic trees constructed with combining of DNA barcoding markers yielded similar grouping, like in the case of individual DNA barcodes (Figure 2), and repeating most of the peculiarities established by using individual DNA barcodes. It is difficult to evaluate whether the grouping of different markers provides better results or not.

Taxonomic assignment of *Thymus* specimens through Basic Local Alignment Search Tool (BLAST) analyses [42] against publicly available accessions in NCBI did not return reliable results, probably because the publicly available sequences in the databases represent mostly well-known and studied species with a large distribution, and, to a lesser extent, species from insufficiently studied regions (at least by using molecular markers), such as the Balkans and Bulgaria in particular. Therefore, the most similar to Bulgarian species were ones with a remote distribution, such as *T. japonicus* (H. Hara) Kitag., *T. mongolicus* (Ronniger) Ronniger (rbcL barcoding region) and for the ITS region, *T. quinquecostatus* Celak. and *T. serpillum* L. (results not shown). Evidently, much more information and richer databases are necessary for reliable application of the BLAST analysis to the Bulgarian *Thymus* species.

As discussed by [1], often in TCGs, the fragmented populations of the species with reduced or lacking gene flow and long-term evolution in isolation result in spatial structures with morphological and genetic divergence without a strong correspondence among each other. This makes distinguishing taxa very difficult without clearly determined taxonomic boundaries. Plant taxonomy is a complex issue, and speciation processes could be extremely variable, especially in TCGs [43,44,45].

We consider the present study as a first step toward an updated and taxonomically sound classification of the Bulgarian species of the genus *Thymus*. Including the remaining indigenous species and such that are not occurring naturally in the country but are important as key species with well-known and established taxonomic positions could facilitate the process of obtaining a proper and reliable classification.

## 3. Material and Methods

### 3.1. Plant Material

Fourteen samples were included in the study, representing eleven species. Five species belong to the section *Hyphodromi* (A. Kerner) Halácsy and six species to the section *Serpillum* (Miller) Bentham. The sample set represents all sections of the genus found in Bulgaria. *Thymus longedentatus*, *T. perinicus* and *T. stojanovii* were represented by two samples, while the other species were represented by only one sample (Table 3). The taxonomic system of the genus used in the present study is generally the same accepted in [3] and in Euro+Med PlantBase (https://ww2.bgbm.org/EuroPlusMed/; last accessed 23 December 2021), with slight modifications by [8]. The differences due to the modifications concern only a few taxa: for example, in [8], *T. degenii* H. Braun is treated as part of *T. sibthorpii* Benth.; *T. kosteleckyanus* Opiz as part of *T. pannonicus* All.; *T. odoratissimus* Mill. as a synonym of *T. glabrescens* Willd. However, none of the taxa differing in their taxonomic status between [3] and [8] were included in the present study.

### 3.2. DNA Extraction, PCR Amplification and Sequencing

Leaves dried in silica gel (~30 mg) were ground to powder by a Mixer Mill MM 40 (Retsch GmbH, Haan, Germany). Genomic DNA was extracted by using an Invisorb^®^ Spin Plant Mini Kit (Invitek Molecular GmbH, Berlin, Germany). DNA concentration and purity was measured by a NandropTM Lite Spectrophotometer (Thermo Fisher Scientific, Waltham, MA, USA). The genetic diversity of samples was evaluated based on sequences of universal barcodes for plants psbA-trnH, matK, rbcL and ITS. The sequence of used primers (synthesized by Microsynth) and PCR conditions that varied among primers are shown in Table 4. PCR amplification was performed in 20 µL reaction mixtures containing approximately 30 ng of genomic DNA, 1 x PCR buffer, MgCl_2_ (2.0 mM for ITS and matK, 2.5 mM for rbcL and trnH-psbA), 0.2 mM of each dNTP, 0.2 µM of each primer and 1.0 U Taq DNA Polymerase (Solis BioDyne, Tartu, Estonia). The quality of PCR products was checked on 1% agarose gel containing GoodViewTM staining dye. Successful amplicon products were sequenced in both directions by Microsynth (Göttingen, Germany) using the same primers used for PCR amplification.

### 3.3. Sequence Alignment and Data Analysis

Candidate DNA barcodes sequences for each barcode region were edited and aligned in the software packages Geneious (Geneious Prime 2022.0.1) or MEGA-X [46], and consensus sequences were subjected to further analyses. The phylogenetic trees were constructed using the Geneious software testing genetic distance models provided by the software package—the Jukes–Cantor model [47], Hasegawa–Kishino–Yano (HKY) model [48] and Tamura–Nei model [49]. The statistical parameters of genetic diversity (total number of sites, number of variable sites, number of parsimony informative sites singleton sites) were calculated for each DNA barcode marker and some of their combinations in the MEGA-X software package.

## 4. Conclusions

Low bootstrap support testifies to the unreliability of the majority of groups identified on phylogenetic trees and casts doubt on the possibility of using the studied markers to study phylogenetic relationships in a taxonomically complex group, such as the genus *Thymus*. A weak trend in pooling samples of the same species indicates the low value of the studied markers for barcoding and suggests the need to look for other markers. Future approaches to the study of *Thymus* taxonomy and phylogeny should be more complex, including a set of molecular, morphological and other phenotypic markers. Additionally, population genetic studies could provide a reliable picture of the distribution of genetic diversity and the degree of differentiation and could help delineate the real distinct populations and, possibly, the natural entities at the species level, as suggested by [1].

## Figures and Tables

**Figure 1 plants-11-00270-f001:**
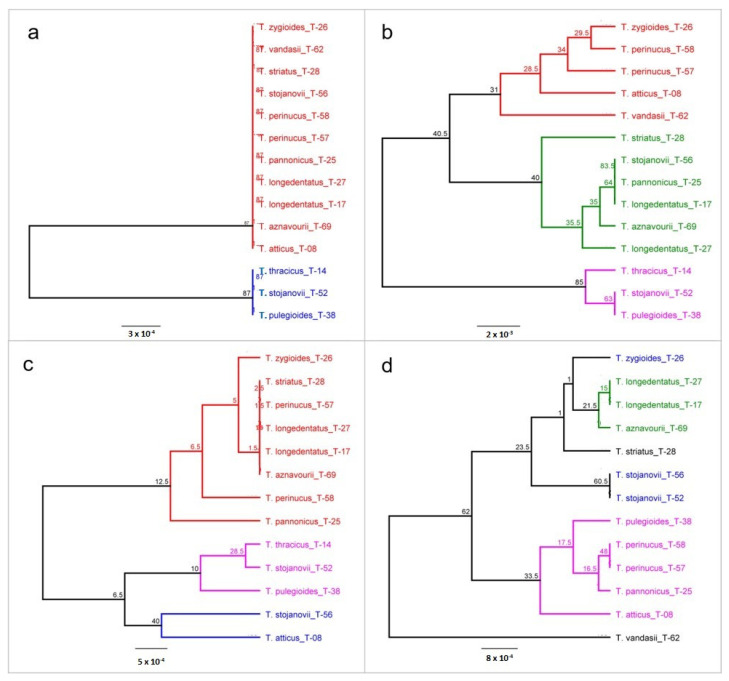
Phylogenetic trees of *Thymus* taxa constructed based on barcode regions rbcL (**a**), trnH-psbA (**b**), matK (**c**) and ITS (**d**). The trees were constructed using the Geneious software with the genetic distance model Jukes–Cantor, the unweighted pair group method with averages UPGMA clustering method and the resampling method bootstrap with 200 replicates. Bootstrap values > 50% are shown along the branches.

**Figure 2 plants-11-00270-f002:**
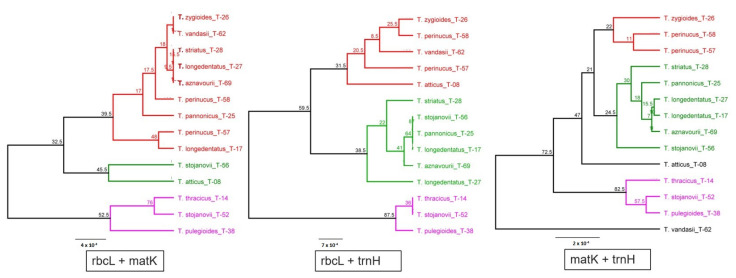
Consensus phylogenetic trees of *Thymus* taxa constructed based on a combination of plastid DNA barcode markers. The trees were constructed using the Geneious software with the genetic distance model Jukes–Cantor, the clustering method UPGMA and the resampling method bootstrap with 200 replicates. Bootstrap values > 50% are shown along the branches.

**Table 1 plants-11-00270-t001:** Efficiency of PCR amplification and sequencing for *Thymus* accessions for four DNA barcode regions.

Barcode Region	N (Samples Tested)	Alignment Length (bp)	Percentage of Amplification Efficiency	Percentage of Sequencing Efficiency (from Amplified Barcodes)
matK	14	760	100	100
rbcL	14	530	100	100
trnH-psbA	14	350	100	100
ITS	14	619	93.4	93.4

**Table 2 plants-11-00270-t002:** Statistical parameters of genetic diversity calculated in MEGA X.

DNA Barcode Region	Ns	C	V	Pi	S	Average Pairwise Distance/SE
rbcL	529	527	2	2	0	0.00127/0.00009
trnH-psbA	351	333	16	10	6	0.13205/0.00371
matK	761	748	13	3	10	0.00342/0.00171
ITS	618	604	14	2	12	0.00464/0.00144
rbcL+matK	1290	1275	15	5	10	0.00230/0.00066
rbcL+trnH-psbA	880	860	18	12	6	0.00577/0.00147
matK+trnH-psbA	1112	1081	29	13	16	0.00748/0.00176

Legend: Ns—total number of sites; C—number of constant sites; V—number of variable sites; Pi—number of parsimony informative sites; S—singleton sites. Estimates of Average Evolutionary Divergence over all Sequence Pairs from the number of base substitutions per site are shown. The standard error estimate(s) were obtained by a bootstrap procedure (1000 replicates). Analyses were conducted using the Tamura–Nei model (see Material and methods). The rate variation among sites was modeled with a gamma distribution (shape parameter = 1). All ambiguous positions were removed for each sequence pair (pairwise deletion option).

**Table 3 plants-11-00270-t003:** List of the studied specimens with details of their geographic locations.

Species	Sample Code	Geographic Coordinates Altitude
Section *Hyphodromi* (A. Kerner) Halácsy		
Subsection *Subbracteati* (Klokov) Jalas		
*Thymus atticus* Čelak.	T08	41°25′46″ N 23°42′42″ E 760 m
*Thymus perinicus* (Velen.) Jalas	T57	41°46′13″ N 23°24′28″ E
	T58	2450 m
*Thymus striatus* Vahl.	T28	41°57′52″ N 22°56′10″ E 900 m

Subsection *Serpyllastrum* Huguet del Villar		
*Thymus aznavourii* Velen.	T69	41°50′04″ N 26°28′55″ E 120 m
*Thymus zygioides* Griseb.	T26	41°39′07″ N 25°50′39″ E 300 m
Section *Serpyllum* (Miller) Bentham		
Subsection *Isolepides* (Borbás) Halácsy		
*Thymus longedentatus* (Deg. et Urum.) Ronniger	T17	42° 03′24″ N 24°26′17″ E 300 m
	T27	41°40′30″ N 25°49′57″ E 480 m
*Thymus pannonicus* All.	T25	41°39′06″ N 25°50′39″ E 300 m

Subsection Alternantes Klokov		
*Thymus pulegioides* L.	T38	41°47′ 07″ N 23°27′40″ E 1500 m
Subsection *Pseudomarginati* (Braun ex Borbás) Jalas		
*Thymus stojanovii* Deg.	T52	41°24′37″ N 23°38′57″ E 1600 m
	T56	41°33′04″ N 24°25′46″ E 1300 m
*Thymus thracicus* Velen.	T14	41°46′14″ N 23°24′48″ E 2200 m
*Thymus vandasii* Velen.	T62	42°12′18″ N 23°19′20″ E 2200 m

**Table 4 plants-11-00270-t004:** Oligonucleotide primers used and PCR conditions.

Barcode Region	Primers	Primer Sequences 5′-3′	PCR Conditions
matK	MatK-RKIM-f	ACCCAGTCCATCTGGAAATCTTGGTTC	95 °C 5 min 95 °C 30 s, 51 °C 50 s 72 °C 1.4 min, 35 cycles 72 °C 7 min
	MatK-3FKIM-r	CGTACAGTACTTTTGTGTTTACGAG	
rbcL	rbcLa-F	ATGTCACCACAAACAGAGACTAAAGC	94 °C 4 min 94 °C 30 s 55 °C 30 s 72 °C 1 min, 35 cycles 72 °C 10 min
	rbcLajf634R	GAAACGGTCTCTCCAACGCAT	
trnH-psbA	psbA-trnH	CGCGCATGGTGGATTCACAATCC	94 °C 4 min 94 °C 30 s, 55 °C 30 s, 72 °C 1 min, 35 cycles 72 °C 7 min
	psbA-3F	GTTATGCATGAACGTAATGCTC	
ITS	ITS_F1	CCTTATCATTTAGAGGAAGGAG	94 °C 5 min 94 °C 30 s, 50 °C 30 s, 72 °C 1min, 35 cycles 72 °C 5 min
	ITS 4	TCCTCCGCTTATTGATATGC	

## Data Availability

Data are available from authors on request.

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
