# Peer review of "DNA Barcoding Study of Representative Thymus Species in Bulgaria"

_plants, 2022, doi:10.3390/plants11030270_

Round 1
Reviewer 1 Report
The present manuscript is the first molecular phylogenetic study of the genus Thymus from Bulgaria. Thymus is a genus poorly studied by molecular methods. Previous studies of the genus showed rather weak phylogenetic signal in molecular data. Testing the markers for their relevance for the separation of the Bulgarian Thymus species was initially not very promising, but the authors nevertheless made this attempt.
The study includes fourteen specimens from eleven species of Thymus, representing two sections. In the introduction it would be better to add a brief description of the taxonomic system of the genus Thymus adopted in this study. Then it is necessary to indicate which subgenera and sections are represented in Bulgaria and which of them were included in the study. It is desirable that at least all sections of the genus found in Bulgaria be represented in the work. The introduction does not sufficiently describe previous molecular studies of the genus Thymus and the tribe Mentheae.
The methods are described very briefly. The authors did not explain the choice of methods. Why weren't other methods of phylogenetic reconstruction applied, such as maximum likelihood analysis or Bayesian analysis? Why was the Jukes-Cantor model chosen? Has a model test been performed? How did you deal with gap-rich positions? Based on the results obtained for the individual markers, combining the markers is unlikely to improve the resolution, but such an attempt should have been made, at least for plastid markers.
In the results, the authors did not provide some important results, such as statistical parameters of genetic diversity for each studied marker (total number of sites, number of variable (polymorphic) sites, number of parsimony informative sites etc.). The authors do not provide bootstrap support for nodes. But if the clades that stand out on phylogenetic trees are not supported, their isolation is generally questionable. The result that plastid markers vary independently of each other is very important. This demonstrates the need to study the evolution of plastid genomes in taxonomically complex groups and species complexes.
Overall, the results are interesting and worthy of publication, but the study and manuscript need significant improvement.
Some comments are given in the pdf-file.

Author Response
Reviewer 1.
- The study includes fourteen specimens from eleven species of Thymus, representing two sections. In the introduction it would be better to add a brief description of the taxonomic system of the genus Thymus adopted in this study.
Answer: Done – A paragraph was added indicating that the taxonomic system.
- Then it is necessary to indicate which subgenera and sections are represented in Bulgaria and which of them were included in the study. It is desirable that at least all sections of the genus found in Bulgaria be represented in the work.
Answer: Done – we underlined that although the study includes 11 of the species naturally occurring in Bulgaria, they represent all sections of the genus found in the country.
- The introduction does not sufficiently describe previous molecular studies of the genus Thymus and the tribe Mentheae.
Answer: The introduction was updated with reviewing more molecular studies of the genus Thymus and the tribe Mentheae.
- The methods are described very briefly. The authors did not explain the choice of methods. Why weren't other methods of phylogenetic reconstruction applied, such as maximum likelihood analysis or Bayesian analysis? Why was the Jukes-Cantor model chosen? Has a model test been performed? How did you deal with gap-rich positions?
Answer: More extensive description of the method was provided.
- Based on the results obtained for the individual markers, combining the markers is unlikely to improve the resolution, but such an attempt should have been made, at least for plastid markers.
Answer: We applied combinations of the plastid markers, and as expected by the reviewer, the combinations did not improve the results.
- In the results, the authors did not provide some important results, such as statistical parameters of genetic diversity for each studied marker (total number of sites, number of variable (polymorphic) sites, number of parsimony informative sites etc.). The authors do not provide bootstrap support for nodes. But if the clades that stand out on phylogenetic trees are not supported, their isolation is generally questionable.
Answer: A table was inserted in the text with the main statistical parameters. Also, bootstrap was applied in constructing the phylogenetic trees.
Reviewer 2 Report
The manuscript entitled "DNA barcoding study of representative Thymus species in Bulgaria" tested the effectiveness as taxonomic and phylogenetic discriminant markers of four DNA barcording standard genes.
The manuscript is overall correctly structured and well written, in a correct English form,some misprints are directly highlighted in the text.
Each section is properly conceived and argued, but some aspects still need some further insights.
First of all, plant sampling seems to be too strict, not in terms of investigated taxa, but of analysed individuals per taxon. Just one sample for most species, except two, is really poor in order to define a DNA barcorde at specific level. No specification was provided about the sampling choice.
Furthermore, given that the analyses on each single marker did not provide significant resolutions, it is not clear to me why the authors did not analyse multiple combined alignements and verify the resulting tree topologies, instead simply saying that this procedure often does not produce significant results (cfr. pag. 5 , lines 141-142).
Overall, in my opinion, the manuscript can be eligible for publication after the above mentioned improvements.

Author Response
- First of all, plant sampling seems to be too strict, not in terms of investigated taxa, but of analysed individuals per taxon. Just one sample for most species, except two, is really poor in order to define a DNA barcode at specific level. No specification was provided about the sampling choice.
Answer: We agree that only one individual per species/taxon is not sufficient if we wish to define a DNA barcode. However, this is just a pilot just a pilot study and the objective was not only to get information about the taxonomy of Thymus species in Bulgaria, but also to check the effectiveness of the DNA barcoding regions.
Furthermore, given that the analyses on each single marker did not provide significant resolutions, it is not clear to me why the authors did not analyse multiple combined alignments and verify the resulting tree topologies, instead simply saying that this procedure often does not produce significant results (cfr. pag. 5 , lines 141-142).
Answer: We added in the study such analyses by combining DNA plastid barcodes – 3 double barcode combinations
Round 2
Reviewer 1 Report
The authors corrected manuscript according to reviewer recommendations. The manuscript can be accepted for publishing, but after minor revision.
I would recommend that the authors correct the following two points:
1) Please, remove bootstrap support from single branches on Figures 1 and 2. It is only meaningful for nodes that link two or more branches.
2) I would recommend to add to the conclusion a phrase like this: Low bootstrap support testifies to the unreliability of the majority of groups identified on phylogenetic trees, and casts doubt on the possibility of using the studied markers to study phylogenetic relationships in the genus Thymus. A weak trend in pooling samples of the same species indicates the low value of the studied markers for barcoding and suggests the need to look for other markers.

Author Response
Dear Reviewer,
Thank you so much for your valuable suggestions! We made the corrections on Figure 1 and Figure 2, and added the conclusion part, using your suggestion. Now, the paper looks really better!

Reviewer 2 Report
The manuscript, though maintaining the character of a pilot study, in the current version was certainly improved and contributes to increase the GenBank availability for the genus Thymus. I think it is acceptable for publication in the journal
Author Response
Dear Reviewer,
Thank you so much for your valuable suggestions and positive evaluation! Now, the paper looks really better!